# Maximum Temperature and Solar Radiation as Predictors of Bipolar Patient Admission in an Emergency Psychiatric Ward

**DOI:** 10.3390/ijerph16071140

**Published:** 2019-03-29

**Authors:** Andrea Aguglia, Gianluca Serafini, Andrea Escelsior, Giovanna Canepa, Mario Amore, Giuseppe Maina

**Affiliations:** 1Section of Psychiatry, IRCCS Ospedale Policlinico San Martino, 16132 Genova, Italy; gianluca.serafini@unige.it (G.S.); andrea.escelsior@live.com (A.E.); jovycanepa@libero.it (G.C.); mario.amore@unige.it (M.A.); 2Department of Neuroscience, Rehabilitation, Ophthalmology, Genetics, Maternal and Child Health, University of Genoa, 16132 Genoa, Italy; 3Psychiatric Clinic, “San Luigi Gonzaga Hospital” of Orbassano—“Rita Levi Montalcini” Department of Neuroscience, University of Turin, 10043 Turin, Italy; giuseppemaina@hotmail.com

**Keywords:** bipolar disorder, temperature, emergency psychiatry, meteorological variables, daylight exposure, sunlight

## Abstract

Environmental variables can regulate behavior in healthy subjects. Recently, some authors investigated the role of meteorological variables in bipolar patients with an impact on both the onset and course of bipolar disorder (BD). The aim of this study was to investigate the impact of meteorological variables and other indexes in bipolar hospitalized patients. We examined all patients admitted to the Psychiatric Inpatient Unit of San Luigi Gonzaga Hospital, Orbassano (Turin, Italy) from September 2013 to August 2015, collecting several socio-demographic and clinical characteristics. Seven hundred and thirty patients were included. Compared to the day of admission of control individuals, patients with BD were admitted on a day that presented higher minimum, medium, and maximum temperature, higher maximum humidity, higher solar radiation, and higher hours of sunshine. After logistic regression analysis, admissions to the emergency psychiatric ward due to a primary diagnosis of BD were associated with maximum temperature and solar radiation. The current study provides a novel perspective on the question surrounding seasonal mood patterns in patients with BD. A greater awareness of all possible precipitating factors is needed to inform self-management and psycho-educational programs as well as to improve resilience regarding affective recurrences in the clinical practice.

## 1. Introduction

The timing of specific activities has fitness consequences as, being active in the same context on the tidal, diel, lunar or seasonal scale at different times, exposes both animals and humans to a natural rhythmic environment fluctuating between light and dark, ambient temperatures, humidity, and tides [1]. 

The relation between seasonal changes, meteorological variables, and psychiatric manifestations has been supposed since Hippocrates observed that both affective temperaments and mood of inhabitants of a particular geographical region changed due to environmental factors such as ambient air, weather, and climatic conditions with the consequent introduction of the seasonality concept in major affective disorders [2]. Interestingly, Lombroso emphasized the role of thermometric and barometric variations on individual mental conditions in the second half of the 20th century [3]. The city of Turin has a long tradition in the investigation of the relation between environmental variables and number of medical hospitalizations in patients with mood disorders.

Bipolar disorder (BD) may be recognized as a severe multifactorial psychiatric condition which affects at least 1–2% of the whole population and it is commonly characterized by affective recurrences consisting in rapid alternations of hypo(manic), depressive or mixed episodes [4].

A large amount of evidence indicated that the impaired adaptation to environmental stimuli in BD is linked to biological clock abnormalities, potentially leading to circadian rhythm alterations, and unstable and hypersensitive mood conditioning the disease course [5,6,7,8,9]. 

Circadian rhythms and metabolism are fundamentally linked: variations in physiological and behavioral processes, temperature, hormone secretion, food intake, sleep, mood, and social rhythms are closely related to chronotype preference, which may be defined as the individual’s preferred period of activity reflecting a circadian or ultradian propensity for alertness or somnolence [10,11]. 

In healthy subjects, environmental variables such as sunlight exposure, photoperiod and solar radiation/insolation can regulate behavior affecting circadian functions and detemine changes of many biological rhythms. Recently, some authors investigated the role of these factors in bipolar patients with significant impact on both the onset and course of BD and consequently on sleep-wake cycle, mood and energy [10,12,13,14,15,16]. Furthermore, certain authors indicated that environmental factors, such as humidity, day length, ultraviolet radiation, temperature, rainfall/atmospheric pressure or airborne allergens exert a significant role on the admission of BD patients, particularly in (hypo)manic phase, or when they exhibit specific symptoms such as psychosis, impulsiveness, and suicidal behavior/aggression [14,17,18,19,20,21,22,23,24,25]. 

However, specific environmental and meteorological factors responsible for the onset of an affective episode or recurrences in BD are still poorly understood, although it is well recognized that a significant proportion of BD patients showed a seasonal vulnerability, as described in Diagnostic and Statistical Manual of Mental Disorders, fifth edition (DSM 5) within the specifier criteria of BD [18,21]. Thus, exploring in a more detailed manner the environmental factors associated with BD admissions might guide early intervention and prevention strategies towards distressing mood episodes.

Given this background, the aim of the present study was to investigate the impact of meteorological variables in a large sample of hospitalized patients with a primary diagnosis of BD and a control group including individuals admitted in the same time frame for principal diagnoses other than BD.

We mainly hypothesized a significant association between the amount of bipolar subjects’ admissions and the mean sunshine hours. In addition, we hypothesized that specific meteorological factors such as maximum temperature and solar radiation may significantly affect the admissions of bipolar patients in the emergency psychiatric ward, first according to the higher number of studies looking at the association with medical conditions and secondly based on the recent increasing amount of literature focusing on meteorological variables, particularly temperature, and mental disorders [26,27,28].

## 2. Materials and Methods

### 2.1. Sample

We initially evaluated about 900 patients from Psychiatric Inpatient Unit of San Luigi Gonzaga Hospital, Orbassano (University of Turin) over a 24-month period from 1st September 2013 to 31st August 2015; however, only 730 subjects voluntarily accepted to participate in the study. 

This is a referral center that is mainly responsible for patients from the Local Health District (Collegno-Orbassano). All patients consecutively recruited in the present study were subjects from northwest Italy. All re-admitted patients within two months by the previous discharge were excluded from the study to avoid both the so-called “revolving door patients” and patients during the same illness episode.

Psychiatric admission in Italy is regulated by a mental-health law issued in 1980. Involuntary admission occurs when the three following conditions were satisfied: (a) the patient experiences mental changes that require an urgent therapeutic intervention; (b) the patient does not accept treatment; and (c) there are no conditions enabling to take other timely and adequate therapeutic measures outside those achieved in hospitals. During the hospitalization, the patients’ status may shift from involuntary to voluntary; specifically, after seven days, a reassessment is usually required [29]. 

Patients voluntarily participated in the study and provided their informed consent (they were not under involuntary psychiatric surveillance when recruited). The study design was conducted in accordance with the guidelines provided in the current version of the Declaration of Helsinki. The present study was approved by the local Ethical Review Board.

### 2.2. Assessment

Basic socio-demographic and clinical features were obtained through the administration of a semi-structured interview used in previous studies with a format that covered the following areas: socio-demographic characteristics such as age, gender, marital, and occupational status, education level, clinical features such as age of illness onset and length of psychiatric illness, month upon entry in the psychiatric ward, current suicidal behavior, type and reason related to admission, overall length of hospitalization, calculating the difference between the admission and discharge day.

All patients were diagnosed according to DSM 5 [30]. All participants, who were recruited directly from our catchment area and have been involuntary/voluntary admitted to our inpatient service, were assessed only when their psychopathological conditions were considered clinically stable by two psychiatrists. If patients were affected by more than one psychiatric diagnosis, only the primary condition was recorded, due to the pharmacological treatment prescribed by a senior psychiatrist (with at least ten years of clinical experience in the field).

Meteorological data were obtained by the Italian Meteorology’s Climate Data Service of Physics Department of the University of Turin (Latitude: 45°03′07,15″ Nord, Longitude: 007°40′53,30″ Est, Altitude: 254 m above sea level) [31] for the day of admission. In particular, daily data pertaining to medium and maximum wind (measured in km/h), minimum, medium and maximum temperature (measured in degrees Celsius), humidity (measured in percentage), and barometric pressure (measured in hPa), solar radiation (measured in W/m^2^), rain (measured in mm), and hours of sunshine were retrospectively matched by date to the primary diagnosis. 

Wind chill index and humidex (humidity index) were also measured. The first one is defined as the perceived decrease in air temperature experienced by the body on exposed skin due to the flow of air while the second one describes how hot the weather is perceived by the average person, by combining the effect of heat and humidity, respectively.

### 2.3. Statistical Analyses

We carried out all the statistical analyses using the Statistical Package for Social Sciences (Version 21.0, SPSS; SPSS Inc., Chicago, IL, USA) for Windows 21.0 with significance which was set at *p* < 0.05 (two-tailed). 

Continuous variables were represented as mean and standard deviation (*SD*) while categorical variables were represented as frequency and percentage considering socio-demographic and clinical characteristics. The total sample was divided in two subgroups: the first one represented by patients with BD, and the second one by the patients receiving other (other than BD) psychiatric diagnoses, respectively. 

Data were analyzed with the Student’s *t*-tests/ANOVA and Pearson’s chi-square test in contingency tables (chi-squared). The Kolmogorov–Smirnov test was conducted to confirm whether all the investigated sample variables followed the normal distribution. A logistic regression analysis was used to identify the most significant socio-demographic, clinical and meteorological correlates to bipolar patients’ admission in an emergency psychiatric ward. The probability of entering the equation was set at 0.05.

## 3. Results

A total of seven-hundred-thirty admissions to the psychiatric unit were recruited over a 24-month period for the present study. 

The mean (±SD) age of the sample was 43.4 (±13.9) years; 311 patients (42.6%) were females; more than half of participants (55.6%) were single, and about a third of the total sample was employed at the time of recruitment. Concerning clinical characteristics, the mean age of onset was 28.5 (±13.3) years, while the mean duration of hospitalization was 11.4 (±8.9) days. Overall, one-hundred-twelve (*n* = 112, 15.3%) patients were involuntarily admitted. 

The grouped diagnostic categories are displayed in Table 1, patients with a primary diagnosis of bipolar disorders were 251 (34.4%) of the total sample. 

The most relevant socio-demographic and clinical characteristics of the included subjects are summarized in Table 1.

The sample was then divided in two subgroups: one for patients with BD (*n* = 251, 34.4%), and one for individuals with other psychiatric diagnoses (*n* = 479, 65.6%). Figure 1 shows the monthly admission prevalence according to longitudinal diagnosis. The hospitalization of bipolar patients showed significant peaks during the months with more sunlight exposure.

When the two subgroups were compared, as shown in Table 2, patients with BD were admitted on a day that presented, as compared to the day of admission of control individuals, higher minimum, medium, and maximum temperature, higher maximum humidity, higher solar radiation, and higher hours of sunshine. 

Furthermore, both humidex and wind chill indexes showed the higher average score in thesubgroup of bipolar patients (Table 2).

When we performed a logistic regression analysis, admissions on emergency psychiatric ward due to a primary diagnosis of BD were associated with maximum temperature and solar radiation (Table 3).

## 4. Discussion

Our findings suggested that, when compared to the day of admission of control individuals, bipolar patients were admitted on a day that presented higher minimum, medium, and maximum temperature, higher maximum humidity, higher solar radiation, and higher hours of sunshine. It is well known that bad weather makes us sad while good weather makes us happy with this meteorological factor exerting a direct impact on our mood. Extreme weather changes, caused by climatic warming, strongly affect global wellbeing as well as the entire eco-environment. According to the current literature, there is increasing research and clinical interest focused on the impact of environmental factors on psychiatric conditions, especially in patients with a significant genetic vulnerability such as those with BD. 

Specifically, over the last decade, a number of researchers thoroughly explored this phenomenon assuming that meteorological variations affect not only human behavior, but also the onset and course of major psychiatric disorders such as BD [12,13,15,32]. A significant link between seasonality and affective recurrences has been reported in BD. Based on recent studies [21,33,34], manic episodes commonly peak during spring/summer and autumn, to a lesser extent, while depressive episodes peak in early winter and summer, and mixed episodes peak in early spring or mid/late summer with greater seasonal fluctuations concerning mood and behavior in bipolar individuals compared to unipolar depressed subjects or healthy controls. 

In line with our hypothesis, the first significant finding consisted in the increased amount of bipolar subjects’ admissions when the mean sunshine hours was longer. This result confirms the role of meteorological variables such as sunshine hours on the admission rates of bipolar subjects when using a different methodology rather that that used in previously published studies [14,16], suggesting that the photoperiod reaches its maximum extension during summer and its minimum during winter in Italy. Importantly, increased sunlight exposure during the day has been hypothesized to play a stabilizing effect on specific neurotransmitters levels such as monoamines that are relevant to mood disorders [6]. Patients with BD present, during acute and euthymic episodes, circadian gene mutations that might lead to a dysregulation of neurotransmitter mechanisms with negative consequences on the normal synchronization of environmental stimuli [35]. Specifically, this altered interaction begins at the level of the supra-chiasmatic nucleus of the hypothalamus (NSI), the nerve fibers of which project both to the raphe nuclei and pineal gland, producing several metabolic, circadian and sleep-wake rhythm dysregulations (due to hypothalamus-pituitary-adrenal gland axis abnormalities) with a final alteration of the immune response and increased oxidative stress at cellular level [6,36,37]. 

Several authors reported a positive association between the intensity of sunlight exposure and hyperthymic temperament [38], the intensity of sunlight exposure and manic/mixed onset of BD [14,16,22,24,25,39,40], although not all studies replicate these findings [17,18,41,42,43,44]. The opposing findings between studies using cross-sectional or prospective data (i.e., hospital admission records or self-report data) might probably be explained according to methodological differences. The comparison of studies, that were carried out at different latitudes and compared samples of populations living in different geographical regions focusing on different meteorological characteristics, is undoubtedly difficult.

The increase in hours of sunshine during spring/summer is also associated with several changes of climatic factors. In our sample, the hospitalization of bipolar patients was closely related with specific weather variables. In particular, we found significant differences between temperature (minimum, maximum, and medium), maximum humidity, solar radiation, humidex and windchill Index, and the day of admission of patients with BD on emergency psychiatric ward compared to the control group.

An increasing amount of findings supports our results confirming the potential role of temperature on the whole number of patients consulting the emergency rooms in first aid [45], hospitalized bipolar patients (regardless of their mood episodes) [17,46], and the (hypo)manic [22,24,47] or depressed patients with affective recurrences [48]. To this aim, Volpe and coworkers evaluated the relation between climatic factors and three dimensions of mania (e.g., suicidality, psychosis and aggression) in 425 bipolar patients, showing a significant association with increasing temperatures for suicidal behavior, increasing hour of sunshine and hours of sunshine of index month for psychosis [19]. However, other studies did not find any association between BD and climatic factors [18,42]. The lack of consensus on the weather–mood paradigm in current literature focused on BD is presumably due to variations in how mood disorder states are measured and conceptualized in BD (e.g., episodes, hospitalization, self-reports) and the most appropriate time lag to monitor this hypothesized relation (synchronous, previous day, weekly, monthly). Unfortunately, we were unable to establish a causal relation between the examined variables due to their strong interdependence and the fact that meteorological variables are mutually dependent on each other (e.g., sunshine and UV dose are highly positively inter-correlated as well as sunshine, temperature and humidity or temperature and humidity and/or windchill index). Therefore, this finding prompted the following question: is hours of sunshine/sunlight exposure really the most incisive environmental variable for assessing affective recurrences in patients with BD, or are there other meteorological factors predominant instead? 

Aiming to correctly address this question, we performed a logistic regression analysis and, in line with our hypotheses, we found that maximum temperature and solar radiation were the most important and significant meteorological factors associated with the admissions of bipolar patients in the emergency psychiatric ward. 

The lack of significance of other specific variables such as humidex and windchill index might be explained given the close correlation of these indexes with temperature. Indeed, the humidex index describes the perceived warm by combining the effects of air temperature with those of relative humidity while the windchill index measures the skin perceived temperature relative to the wind ability to remove heat from the human body; thus, the latter represents a measure of the rate of heat lost by the body. The lack of significance of minimum/mean temperature is presumably due to the fact that only maximum temperature is closely implicated in determining the onset of mood episodes requiring admission to hospital. Similarly to our findings, Shapira and colleagues found that, on a large sample of depressed patients, the admissions rates of bipolar patients were correlated with maximum environmental temperature [48]. Furthermore, a recent study by Sung et al., using a national cohort of psychiatric inpatients, reported a positive relation between ambient temperature and BD. The authors indicated that the risk of bipolar admissions increased with temperatures over 24.0 °C with heat playing an important role in exacerbating affective episode recurrences of BD, especially during extreme heat conditions. Furthermore, the highest daily diurnal temperatures above 30.7 °C (99th %) had the greatest risks of hospitalizations for bipolar subjects [46]. More recently, a very large study conducted on 24313 manic patients showed that higher admission rates were associated with more ultraviolet radiation and higher temperature [22].

Recently, another important study, conducted on eleven BD type I patients by Bullock and colleagues, stressed the importance of maximum temperature to predict the transition from depressed to manic mood on consecutive days, particularly in the absence of a direct sunshine hours effect on mood. The authors showed that, in a small sample of bipolar patients, daily maximum temperature predicted a clinically relevant mood change, in particular a transition into manic mood states [23]. Despite the fact that almost all cerebral processes are sensitive to temperature fluctuations, which were reported to intrinsically modulate behavioral changes and reflexively generate autonomic responses [49], the authors suggested an active involvement of maximum temperature in a model in which its effect was likely mediated by sleep disruption [23].

One may speculate that the link between the two variables should be interpreted by attributing a more directly causative role to the increased environmental temperature rather than sunshine hours on mood switches. By a neurobiological perspective, the major integrative centre for human thermoregulation is located in the preoptic region and anterior hypothalamus, involving several and complicated mechanisms of neurotransmission. In this framework, it is not surprising that lithium exerts relevant chronobiological effects, acting as a synchronizer and stabilizer of circadian rhythms. Importantly, lithium response is able to influence light sensitivity and melatonin secretion associated with changes in the expression of core clock genes, delays sleep-wakefulness rhythms as well as the peak elevation of diurnal cycle body temperature, reduce amplitude and duration of activity rhythms and increase free-running rhythms, but even induce better sleep efficiency and longer sleep duration even in euthymic patients [5,50,51,52,53].

### Limitations

The general readership should be mindful of a number of limitations in the current study when interpreting our findings. First, other environmental/psychological factors that might contribute to the onset of an acute clinical event (e.g., stressful life events, general medical conditions, poor adherence to treatment, and holidays of caregivers meaning a poor social support) have not been considered and could not be ruled out as possible contributing factors in our findings. Second, our data are limited to a single hospital, and the control group was represented by a mixed sample of psychiatric diagnoses. Third, our results were obtained by patients requiring hospitalization and their seasonal patterns might significantly differ from those not hospitalized such as outpatients who were not included in this study. Furthermore, the effect of other clinical variables such as number of previous hospitalizations, predominant polarity, a structured interview for diagnoses, inability to examine the eventual differences between climatic factors and mood (e.g., manic, hypomanic, depressive, and mixed) states as well as changes of climate days before the admission has been not investigated. Therefore, the generalization from our results should be carried out with caution. Another limitation is the cross-sectional study design of meteorological variables on a single day instead of variations/fluctuations over time, which does not allow inferences on the temporal relation between the variables and only shows measures of association. Furthermore, the lag-time has been not considered. Lastly, meteorological variables measured somewhere outside may not mirror the temperature the patient is exposed to, due to the increasing number of air conditioning systems. This may lead to the well-known ecological regression bias.

## 5. Conclusions

Notwithstanding these limitations, the outcomes of the current study provide a novel perspective on the questions surrounding seasonal mood patterns in patients with BD. We were particularly interested in specific meteorological variables (if any) that could have affected hospitalization in emergency psychiatric wards. Overall, we found a significant variation concerning admission rates correlated with certain weather variables (i.e., particularly maximum temperature and solar radiation) for patients with a primary diagnosis of BD, when compared with subjects to other psychiatric diagnoses. A greater awareness of all possible precipitating factors is needed to inform self-management and psycho-educational programs aimed to improve resilience regarding affective recurrences in the clinical practice, even if these results should be not generalized as proper effects on patients with BD. 

Psychiatrists, both individually and collectively, may have an important role in carefully addressing environmental changes and its relation with psychiatric conditions. Future longitudinal studies should replicate the present findings using larger samples. 

## Figures and Tables

**Figure 1 ijerph-16-01140-f001:**
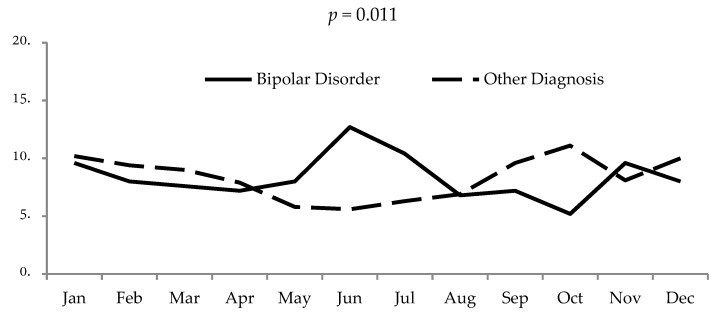
Inpatient admission rates by month: comparison between bipolar and other mental disorders (adapted by Aguglia et al., 2018) [16].

**Table 1 ijerph-16-01140-t001:** Socio-demographic and clinical characteristics in the total sample (*n* = 730).

Characteristics of the 730 Patients Included	Total Sample
Gender (female), *n* (%)	311 (42.6)
Age (years), mean ± *SD*	43.4 ± 13.9
Education level, *n* (%)	
Elementary	67 (9.2)
Low school	352 (48.2)
High school	257 (35.2)
Graduation	54 (7.4)
Marital status, *n* (%)	
Single	406 (55.6)
Married	190 (26.0)
Divorced	106 (14.5)
Widowed	28 (3.9)
Working status, *n* (%)	245 (33.6)
Age at onset (years), mean ± *SD*	28.5 ± 13.3
Suicide, *n* (%)	
Ideation	122 (16.7)
Attempt	77 (10.5)
Admission, *n* (%)	
Involuntary	112 (15.3)
Voluntary	618 (84.7)
Length of stay, mean ± *SD*	11.4 ± 8.9
Diagnosis, *n* (%)	
Bipolar and related disorders	251 (34.4)
Schizophrenia and related disorders	192 (26.3)
Depressive disorders	134 (18.3)
Others	153 (21.0)

*SD*: standard deviation.

**Table 2 ijerph-16-01140-t002:** Difference on meteorological variables in terms of admission in an emergency psychiatric ward between bipolar disorder (BD) and other diagnosis.

Meteorological Variables	Bipolar Disorder’ Admission (*n* = 251)	Other Diagnoses’ Admission (*n* = 479)	*p*
Temperature minimum (Celsius), mean ± *SD*	10.9 ± 7.4	9.5 ± 7.0	0.010
Temperature maximum (Celsius), mean ± *SD*	20.2 ± 9.0	17.7 ± 8.0	<0.001
Temperature medium (Celsius), mean ± *SD*	15.2 ± 8.1	13.3 ± 7.5	0.002
Humidity minimum (%), mean ± *SD*	50.7 ± 16.2	51.2 ± 17.3	0.719
Humidity maximum (%), mean ± *SD*	86.5 ± 7.6	85.1 ± 8.8	0.027
Humidity medium (%), mean ± *SD*	69.3 ± 12.9	70.5 ± 12.5	0.238
Wind maximum (km/h), mean ± *SD*	14.7 ± 5.9	14.6 ± 5.6	0.866
Wind medium (km/h), mean ± *SD*	4.4 ± 1.6	4.2 ± 1.5	0.136
Atmospheric Pressure minimum (hPa), mean ± *SD*	1012.6 ± 8.0	1013.5 ± 8.1	0.136
Atmospheric Pressure maximum (hPa), mean ± *SD*	1018.2 ± 7.2	1019.1 ± 7.7	0.125
Atmospheric Pressure medium (hPa), mean ± *SD*	1015.4 ± 7.2	1016.3 ± 7.9	0.172
Rain (mm), mean ± *SD*	2.6 ± 6.3	2.9 ± 7.7	0.574
*Solar Radiation (W/m^2^), mean ± *SD*	6.9 ± 3.3	5.8 ± 3.2	<0.001
Humidex Index (Celsius), mean ± *SD*	21.7 ± 10.1	19.3 ± 8.9	0.001
Windchill Index (Celsius), mean ± *SD*	9.4 ± 8.3	7.9 ± 7.8	0.017
Hours of sunshine, mean ± *SD*	12.1 ± 2.5	11.6 ± 2.3	0.030

* increase 100 Watt unit.

**Table 3 ijerph-16-01140-t003:** Relationship between potential explanatory variables and admission of bipolar patients in an emergency psychiatric ward: results from logistic regression analysis.

Variables	*p*	OR	95% CI
Gender	0.085	0.746	0.534–1.042
Age	0.004	1.018	1.006–1.030
Temperature maximum	0.030	1.162	1.011–1.231
* Solar radiation	0.025	1.146	1.017–1.292
Hour of sunshine	0.131	0.843	0.675–1.052
Atmospheric Pressure maximum	0.103	0.980	0.957–1.004
Windchill index	0.974	0.999	0.943–1.058
Humidex index	0.300	0.950	0.863–1.046
Humidity maximum	0.180	0.985	0.964–1.007
Lenght of stay	<0.001	1.066	1.044–1.088
Spring	0.354	1.462	0.655–3.263
Summer	0.065	2.251	0.950–5.332
Autumn	0.528	1.185	0.699–2.011
Winter	0.149	0.962	0.901–1.112

* increase 100 Watt units; OR: odds ratio; CI: Confidence interval.

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
