# Peer review of "Maximum Temperature and Solar Radiation as Predictors of Bipolar Patient Admission in an Emergency Psychiatric Ward"

_ijerph, 2019, doi:10.3390/ijerph16071140_

Round 1

Reviewer 1 Report

 Thank you for your reply, the comments are all satisfied.

Reviewer 2 Report

 Thank you for your reply, the comments are all satisfied.

This manuscript is a resubmission of an earlier submission. The following is a list of the peer review reports and author responses from that submission.

Round 1

Reviewer 1 Report

This study about the influence of environmental factors on the onset and progress of bipolar disorder seems to be scientifically performed and it is correctly exposed. The reading clarifies very well how temperature, humidity and solar radiation play an important role in the maniac-depression humoral changes’ in subjects with bipolar disorder. The starting hypothesis is exposed clearly and seems to have a direct association between hours of sunshine, radiation and temperature. The exposure of the results is simply and well-defined, specifically, they suggest that the meteorological factors exert a strong impact on the mood and consequently, have an impact too on the admission number of subjects with bipolar disorders. The admissions increase when the average sun hours were longer, the photoperiod reaches its maximum extent during the summer and its minimum during the winter.

Moreover, the role of the neurobiological area is interesting and well exposed: the increase in hours of sun has stabilizing effects on levels of monoamine neurotransmitters.

The acute and euthymic episodes and circadian dysregulation involves the suprachiasmatic nucleus of the hypothalamus of the metabolic rate and of the immune system.

From a neurobiological perspective, the main integration center for thermoregulation is found in the preoptic region and in the anterior hypothalamus involving different neurotransmission’s mechanisms.

Following this line, it is not surprising that lithium has significant chronobiological effects, acting as a synchronizer and stabilizer of circadian rhythms. Additionally, the lithium response, that influences the sensitivity to the sunlight and the secretion of melatonin, is associated with changes in the expression of the “internal clock genes”, in the sleep-wake rhythms and in the increase of body temperature during the day. These parameters are defined by good sleep in quality and quantity.

Regarding the English language and style, they are fine with minor spell check required.

Following the aforementioned considerations, the reviser sustains the publication without further changes.

Reviewer 2 Report

With great interest i read your contribution to BD hospital admissions and meteorological variables, 

line 85, "We initially screened a sample of 900 patients;" 900 patients sampled from where?

line 105:  ethic committee reference (number) is missing

line 143,144: what means "found associated at the univariate analysis"? based on a significant p-value? In general selection of variables in a regression model using univariate pre-testing meanwhile is known to be highly potentially biased. Therefore this selection method does not seeem to be appropriate in this analysis.

Why are socio-demographic variables lines 108,109 not included in the regression model (Table 3)?

Table 2: In my pdf version some words/characters in lthe left column are shortened/hidden

Column named t probably pointing to the t-statistic does not provide addiotional information to p-value and can be deleted

line 171   df=728 can be deleted, does not provide additional information

The same applies for Table 3 columns "T, E.S., Wald". If Exp(B) stands for odds ratio, OR as header is easier to read

Table 3: Showing the Odds ratio for Solar radiation not by one Watt unit increase but for e.g. by 100 Watts would avoid the very small number of 1.002 which is rather difficult to understand for the readership.

table 3: "mean+-SD" can be shifted to title or footnote but should'nt show up in every line

line 237: concerning inclusion of lag-time: Did you try to include other days of measurements than the current one? If not, why?

A simple descriptive seasonal analysis e.g. a table or graph  by week or month would give some insight on seasonality independent from metereological risk factors. 

Regression model: Including strongly correlated min/median/max values of Temperature simultaneously must lead to rather arbitrary p-values since these three correlated variables compete in the estimation process. Therefore additionally models including only one temperature  measurement as a sensitivity analysis should be done/presented in order to show which of these variables is important or not.

line 238: "causal relation" This comment is very true since this type of observational study design 

is not able to throw light on causality at all.

line 247:  "most important and significant meteor. factors" It should be stressed that the p-value for Temp max is 0.034 which is close to the significance level of 5%. Considering the all omnipresent increased Type 1 error in regression modelling due to multiple testing the p-value for Temp max is not impressive at. Still, considering the collinearity in the data set it is not surprising.

line 208: "we found a significant variations"  plural/singular?

line 311-317: This line  sounds a bit contradictory to me and too unspecific. Knowing the effect of max Temperature and avoiding heat would it avoid a BD hospital admission or just postpone?

Should solar exposure  be avoided?

The regression model deals with the effect of meteorological variables on the odds of BD disorders to "non-BD" psychic disorders but not the risk of BD admission. So,the practical consequences are rather limited since avoiding heat I have higher chance getting a non-BD disorder admission then?.

In light of the increasing number of air condition system it should be noted too, that meterological conditions measured somewhere outside may not mirror the temperature the patient is exposed to. This may lead to the well known ecologic regression bias.

Reviewer 3 Report

Re: Max temp and solar radiation as predictors of bipolar patients’ admission

This is a cross-sectional correlation study on linking the meterological meassure with index admission of psychiatric patients in turin, Italy. The results were most consistent with the current data suggesting an assocaition of metrological factors, especially light and temperature, to the mental disorders. However, there are a number of issues that the authors may need to clarify

1.      Subjects: Sampling: was this a consecutive series of psychaitric admission over 2 years study?

2.      Admission: A major problem for linking the meterological data on the day of admission with the psychiatric diagnosis was the fact that most psychiatric admission will have a time lag from the onset/exacerbation of psychiatric symptoms to the final admission. Did the authors have any data on the day of onset/exacerbation of the symptoms? How many of them were of acute onset problems? In addition, the reason for admission to inpatient was always a complex decision including lot of social issues. In other  words, it will depend on the local admission logistic, say for example, did the patients attend the hospital by themselves or be requested to goto hospitals by relatives? Was the mental state intolerable by the relatives, duly affected by the weather?

3.      Seasonality vs period vs day to day variation:

a.       given that the change of mental state may take days to weeks, it will be much more solid and accurate if the admission could be linked to the weather data in a recent period (?1 week) or seasonality?

b.      Seasonality as confounder: was the effect on admission data due to seasonality effect rather tan daily temperature/sunlight/humidity change? may need to control the confounding effect of seasonality

4.      Length of stay: did the weather data affect the length of stay of patients, while controlling the diagnosis?

5.      Lithium and admission: apart from potential circadian influences, weather may also affect durg metabolism in a numbe of ways, especially Lithium which depends on renal clearance. Was the lithium on admission affected by the weather.